# Psychological Health in Intensive Care Unit Health Care Workers after the COVID-19 Pandemic

**DOI:** 10.3390/healthcare10112201

**Published:** 2022-11-02

**Authors:** Valeria Carola, Cristina Vincenzo, Chiara Morale, Valentina Cecchi, Monica Rocco, Giampaolo Nicolais

**Affiliations:** 1Department of Dynamic and Clinical Psychology and Health Studies, “Sapienza” University of Rome, 00185 Rome, Italy; 2Department of Clinical and Surgical Translational Medicine, “Sapienza” University of Rome, 00185 Rome, Italy

**Keywords:** SARS-CoV-2, COVID-19, clinical psychology, K10, perceived stress, intensive care unit, health care workers

## Abstract

Background: Although the COVID-19 pandemic had an impact on the general population, health care workers (HCWs) constituted one of the groups that were most adversely affected by the associated risks, owing to the significant consequences on their mental health. This study examined these psychological effects on HCWs who cared for COVID-19 patients who were admitted to the intensive care unit in an Italian hospital. Methods: Subjects were administered several self-reported questionnaires: Kessler 10 Psychological Distress Scale (K10), Perceived Stress Scale-10 (PSS), Impact of Event Scale Revised (IES-R), and Post-traumatic Growth Inventory (PTGI), as well as two open-ended questions oriented toward understanding their positive and negative emotional experience and differentiating between two phases of the emergency. Results: Overall, 45% of HCWs showed medium-to-high anxiety/depressive symptoms, whereas 60% presented with medium-to-high levels of perceived stress. In addition, 37% of subjects developed symptoms of PTSD and 50% showed post-traumatic growth in the “appreciation of life” and “new possibilities” dimensions. With regard to the open-ended questions, three themes were identified: quality of workplace relationships, sense of emotional-relational competence, and sense of clinical-technical competence. In addition, two macrocategories of responses were identified in the answers: growth and block. Conclusions: The mental health of HCWs who are involved in the front line of COVID-19 was significantly impacted by this experience, showing high levels of post-traumatic stress and anxiety and depressive symptoms more than 1 year after the emergency began. A qualitative analysis of staff experiences can be a useful guide for structuring interventions and prevention.

## 1. Introduction

The COVID-19 pandemic, which began at the start of 2020, has seriously impacted the economics, health, and social activities of the entire world. SARS-CoV-19 is a member of a family of respiratory viruses that can cause mild to moderate illness, from the common cold to severe and life-threatening acute respiratory syndromes [1,2,3,4,5]. Since the identification of the first cases in December 2019, the number of confirmed cases worldwide has amounted to 278 million, including over 5 million deaths [6]. In Italy, in this period, there were 5,730,040 infections and 136,099 deaths [7].

This danger and uncertainty led to the adoption of emergency measures to contain and combat the pandemic, and several studies [8,9,10,11] have converged in highlighting the psychological impact on the general population, including symptoms related to stress, anxiety, and depression. Although the pandemic has affected the entire population, health care workers (HCWs) constitute one of the groups that have been most seriously affected by its associated risks, particularly with regard to their mental health, especially in frontline HCWs (FHCWs) [12,13,14].

The reduced presence of specialized staff, the fight against an unknown virus, the lack of protective medical equipment, and mandatory social isolation are just some of the challenges that they have had to encounter [15]. In addition, the increase in workload, fear for one’s health and that of loved ones, and the need to make ethically complex choices have increased the psychological stress among health professionals [16]. Anxiety/depressive symptomatology [14,16,17] were the most frequently reported effects by health professionals on the front line in assisting patients who have been infected with COVID-19 [14,16,17], amounting to 20% to 72% of health workers who were involved in the first wave in Italian samples [18,19]. Several studies have highlighted sleep disorders, reporting high levels of insomnia [20,21], increased fatigue [16,22], and high stress [17]. 

Owing to the danger, extraordinary nature, and significant impact of this pandemic, it can be considered a traumatic event. Recent studies in samples of health care professionals who care for COVID-19 patients have shown that they are at risk of developing post-traumatic symptoms [23,24,25], and cases of post-traumatic stress disorder (PTSD) have been observed [26,27,28]. ICU workers have reported high levels of stress [29,30,31] and have shown symptoms of PTSD at clinically significant levels [32], even in the long term, although the incidence of PTSD has waned over time [33]. 

Notably, coping with a traumatic event is a potentially predisposing factor for positive change, defined as post-traumatic growth (PTG) [34]. As a result of stressful circumstances, such as working conditions during the pandemic, HCWs can undergo changes in their perception of personal strength, identifying new scenarios and possibilities in life, experiencing spiritual change, and developing a greater appreciation for life and a greater sense of closeness with others [35]. The presence of PTG, however, does not exclude the possibility that PTSD could develop. In fact, PTG and PTSD can coexist [35,36].

Thus, we hypothesized that having served on the front line during the COVID-19 pandemic is a consistent risk factor for the onset of psychological distress and particularly the development of anxious and depressive symptoms in HCWs. Further, we hypothesized that this situation effected post-traumatic responses and PTG. The purpose of this study is to quantitatively investigate the effects of psychological distress caused by having worked in intensive care during the COVID-19 pandemic, with an in-depth analysis of the qualitative dimensions related to post-traumatic effects, intended both as symptoms and as post-traumatic growth.

## 2. Materials and Methods

### 2.1. Participants

Thirty-five COVID-19 intensive care unit (ICU) HCWs were included (9 men, 26 women: 14 medical doctors and 21 nurses). All HCWs worked in the COVID-19 ICU from March 2020 to June 2021 (period #) and from October 2021 to January 2022 (period #2) at Sant’Andrea University Hospital, Rome. HCWs were invited to complete self-rated questionnaires and answer two open-ended questions on their experience in caring for COVID-19 patients in the ICU 4–6 months after its closure (June 2021). 

### 2.2. Psychometric Tools

The administration of the questionnaires was carried out online through a digital questionnaire. 

The Kessler 10 Psychological Distress Scale (K10), Perceived Stress Scale (PSS), Post Traumatic Growth Inventory (PTGI), and Impact of Event Scale Revised (IES-R) questionnaires were completed by health-care workers, 4–6 months after discharge.

Kessler 10 Psychological Distress Scale (K10). The K10 [37] includes 10 items that investigate distress experienced in the last 4 weeks. A validated Italian translation was used [38]. Participants were asked to respond to each item on a five-point Likert scale, where (1) “none of the time”, (2) “a few times”, (3) “some of the time”, (4) “most of the time”, and (5) “all of the time”; if the previous items score “none of the time”, items 3 and 6 are skipped and a score of 1 is given. Consistently with previous validation studies [38,39], health-care workers were considered to have mild distress when they scored between 20 and 24, whereas they were classified as having high perceived distress when their score was between 27 and 40.

Perceived Stress Scale-10 (PSS). The PSS-10 [40] consists of 10 items that measure how much unpredictable, uncontrollable, and overloaded aspects of one’s life are perceived to be. The questionnaire is organized on a five-point Likert scale ranging from 0 (never) to 4 (very often). Participants are asked to indicate how often they have felt or thought a certain way in the past month. Scores ranged from 0 to 40; higher scores indicated greater perceived stress. A composite score between 18 and 26 was considered intermediate perceived stress. Health worker scores between 27 and 40, on the other hand, were considered high perceived stress. The internal reliability of the PSS-10 appears to be adequate [40].

*Post-traumatic Growth Inventory (PTGI).* PTGI [41] consists of 21 items, which measure possible psychological perceived growth after experiencing a traumatic event. The PTGI is organized into five subscales, including relating to others, new possibilities, personal strength, spiritual change, and appreciation of life. We used it in its Italian version [42]. Participants were instructed to respond to each statement using a six-point scale ranging from “I did not experience this change as a result of my crisis” (scored 0) to “I experienced this change to a very great degree as a result of my crisis” (scored 5). Intermediate scores were given for a very small degree (1), a small degree (2), a moderate degree (3), and a great degree (4). The test–retest reliability (alpha) is 0.71 and the internal consistency of the PTGI is strong (0.90).

*The Impact of Event Scale Revised (IES-R)*. IES-R [43] is a self-report scale with 22 items; the questionnaire assesses the current distress in response to a specific traumatic life event. Avoidance, intrusion, and hyperarousal are the three dimensions added in the revised version of the IES. For our study, we used the Italian version [44]. The questionnaire is organized on a four-point Likert scale ranging from 0 (not at all) to 4 (extremely) through intermediate scores 1 (a little bit), 2 (moderately), and 3 (quite a bit). Participants were asked to respond to each item considering the degree of distress in the past 7 days. The subscale scoring is related to the mean of the responses, allowing the user to immediately identify the degree of symptomatology.

### 2.3. Open-Ended Questions

To analyze in detail the PTG construct, the quantitative analysis performed by PTGI was followed by a qualitative approach. Two open-ended questions were presented to the health care staff to examine their experiences in the care of patients who were admitted to the COVID-19 ICU and determine the positive and negative aspects of this experience and any changes in it over time:*“In your opinion, what positive and negative aspects did the experience of providing health care to patients admitted to the COVID-19 intensive care unit during the first phase of the emergency leave you with?”**“In your experience, are there differences between how you experienced the care of COVID patients during the first phase of the emergency and today? Since the reopening of the COVID wards after the summer, do you feel that you approach the work and the relationship with patients and colleagues in the same way or differently than in the first phase?”*

### 2.4. Statistics

Count data were expressed as frequency and percentage. Measurements were described by the mean and standard deviation. One-way analysis of variance (ANOVA) was performed to assess the impact of the type of socio-health working role covered, on anxious-depressive symptoms (from the K10) and stress-related variables (from the PSS). Significant ANOVA (*p* < 0.05) was followed by post-hoc comparisons using Duncan’s test. Finally, the associations between K10, PSS, IES, PTGI scores were analyzed by Pearson’s correlation Statistical analyses were carried out using Statistica, version 12.0 (StatSoft, Tulsa, OK, USA). 

## 3. Results

### 3.1. Evaluation of Anxious/Depressive Symptoms and Perceived Stress Levels

Frequencies of K10 and PSS scores were recorded and analyzed to assess levels of anxiety/depressive symptoms and perceived stress in HCWs. Forty-five percent of HCWs showed intermediate-high anxious/depressive symptomatology, with a (mean ± SD) of 16.85 ± 6.25 (Figure 1). Further, medium-high levels of perceived stress were evidenced in 60% of HCWs (15.51 ± 8.96; Figure 1).

To determine whether the type of socio-health working role was associated with the level of anxious/depressive symptoms and perceived stress, two ANOVAs were performed. Notably, these analyses showed a significant effect of “role” on PSS scores (F_(1,65)_ = 11.91, *p* < 0.001), wherein nurses had significant lower PSS scores than medical doctors (Figure 2). 

### 3.2. Presence of PTSD Symptoms and Long-Term Post-Traumatic Growth

IES-R questionnaire score frequencies were recorded to detect PTSD symptoms. Overall, 37% of subjects had scores above the clinical cutoff (≥33; total IES = 30.00 ± 16.90; Figure 1). The presence of PTG in each of the subscales of the tool was also evaluated. The fraction of HCWs who scored at or below the normative reference sample on each subscale was 50% for “appreciation of life”, 54% for “personal strength”, 66% for “relating to others”, 50% for “new possibilities”, and 87% for “spiritual change” (Figure 3). Descriptive statistics for the IES and PTGI subscales are shown in Table 1.

In order to verify the associations between all of the psychological parameters measured by the four different scales (K10, PSS, IES, and PTGI), Pearson r correlation analyses were performed (Table 2). K10 total score was significantly positively correlated with the following: PSS total score, IES total score, and all IES subscales. PSS total score was significantly positively correlated with the following: K10, IES total score, and IES—Iperarousal. Significant positive correlations were also observed between the following: IES—Avoidance and PTGI—New possibilities; IES—Intrusiveness and PTGI—New possibilities, PTGI—Personal Strengh, and PTGI—Appreciation of Life; IES—Iperarousal, PTGI—New possibilities, and PTGI—Appreciation of Life; and IES—Total score, PTGI—New possibilities, and PTGI—Appreciation of Life.

### 3.3. Qualitative Analysis—Open-Ended Questions

In the light of PTGI data, we then decided to qualitatively examine the perception of PTG in subjects who experienced a traumatic event through open-ended questions. In the qualitative analysis, ICU staff answered two open-ended questions, which were then analyzed independently by five certified psychologists, who identified three main themes:**Quality of relationships in the workplace**, understood as the level of cooperation and support perceived within the team.**Sense of emotional-relational competence**, considered as the perception of being able to relate to and understand the mental states of oneself, patients, and colleagues.**Sense of clinical-technical competence**, defined as the perception of sufficient technical preparation to deal with clinical emergencies in patients,

The team of psychologists identified two macrocategories of responses. GROWTH referred to an experience of growth, in the acquisition and consolidation of skills that pertained to the subject areas and in the improvement of skills between the two reference periods (period #1 and #2). In contrast, BLOCK referred to a loss of competencies and a barrier to their acquisition or regression of such competencies between the two reference periods. Excerpts of the responses of HCWs are presented below.

#### 3.3.1. Quality of Relations in the Workplace:

(GROWTH) 


*“…help from colleagues in passing on knowledge and affection. I did not give up, and I am grateful for the opportunity, because I realized I was capable”*



*“greater work team cohesion”*



*“certainly, our group has become more united, given the exceptional moment”*


(BLOCK)


*“the patient was more anonymous both relationally and objectively”*


#### 3.3.2. Sense of Emotional-Relational Competence:

(GROWTH) 


*“I felt useful to others”*



*“being able to help”*



*“more self-confidence … unlocking of some emotions”*



*“sharing”*



*“awareness that in the darkest and most critical moments, the best comes out”*



*“Upon reopening, I felt more prepared to handle patients but (having an) absolutely unchanged feeling of helplessness/disbelief about death”*


(BLOCK) 


*“having seen a lot of suffering and not being able to cope with it”*



*“claustrophobic nightmares and bouts of crying at the mere memory”*



*“helplessness”*



*“delusions of cleanliness worsened”*



*“seeing death constantly without having time to understand, if there was anything to understand…”*



*“inability, helplessness, and fright. Having seen so many people die—important physical commitment difficult and demotivating”*



*“worsening relationship with death”*


#### 3.3.3. Sense of Clinical-Technical Competence:

(GROWTH) 


*“I learnt more medical knowledge and learnt to cope better with work stress and to give more importance to teamwork” … “to care for a patient for a pathology up to the end of his life”*



*“assisting patients for a pathology hitherto never dealt with, also regarding new procedures”*


(BLOCK) 


*“long and time-consuming course. Lockdown phase was most depressing and difficult, from a medical and health point of view”*


The recurrence of answers that touted the vaccination campaign as an important moment of change in the perception of risk was also noted:


*“The main difference is the distinction between vaccinated and non-vaccinated patients. The so-called no-vax people were the most difficult part in terms of managing the emotional part”*



*“After the vaccination campaign, I live the pandemic with less fear”*



*“In the first phase of caring for COVID-19 patients, I seriously feared for my life, a fear that was resolved once I underwent the vaccination”*


Further, caring for COVID-19 patients emerged as a challenge for medical staff because of its uniqueness compared with the routine in ICUs. Anesthetists reported profound helplessness in the care of COVID-19 patients, wherein for a long period, there was no effective treatment and patients were mostly alert and in pain, generating uncertainty and helplessness that they did not experience frequently in their usual practice.

## 4. Discussion 

The COVID-19 pandemic has significantly impacted the entire population, with regard to health, economics, and social activity. This study examined psychological distress symptomatology and post-traumatic responses (stress/growth) in physicians and nurses who were engaged in frontline COVID-19 care, several months after closure of their unit.

Distress is an important indicator of mental health and it’s useful in exploring the possible presence of anxiety/depressive symptoms. The significant correlation shown between PSS and K10 is in line with other findings [45] and indicates that, the higher the symptoms of distress, the greater the perceived stress. Consistent with our methodology and results, other authors [46] have investigated the relationship between these dimensions through correlation analyses.

Anxiety and depressive symptomatology were intermediate to high in 45% of our sample.

Anxiety/depressive symptomatology occurs at clinically significant levels in health care personnel [17,47,48]. In Italy, anxiety symptomatology varies between 20% and 72% in medical staff, with the highest percentage recorded in regions in which the pandemic had the most significant impact [18,49]. Studies in other Western countries have reported high percentages of anxious/depressive symptomatology—65% in Turkey [50] and 64% in England [51]. The dependence of the impact of anxiety/depressive symptomatology on the severity of the pandemic was replicated in a French study [52]. On a longitudinal level, studies suggest that, despite the passage of time, as the risk of infection decreases and vaccination rates increase, the levels of traumatic stress remain high among HCWs [31]. 

We also reported significantly higher levels of perceived stress and anxiety/depression in physicians compared with nurses, in contrast to other studies, in which hospital nurses showed greater symptoms of anxiety/depression than the rest of the medical staff, even in Italy [17,30,53,54,55]. This discrepancy is notable, but comparisons between samples are complex, owing to differences in the organization of departments and the specific characteristics of the experiences that contribute to stressogenic situations. 

As confirmed in our interviews, medical staff reported difficulties in managing their work during the pandemic as a result of not only the emergent situation and lack of resources, but also a sense of helplessness and uncertainty that was absent with non-COVID-19 patients. Routine work in the ICU involves short hospital stays, in which the aim is to keep critical patients alive and stabilize them before they can be transported to other wards. The peculiar situation of the pandemic necessitated a readjustment of skills and working methods, such that treatments became much less incisive and were associated with a greater uncertainty of the outcome. We hypothesize that the high stress reported by medical staff resulted from an increased sense of responsibility, coupled with greater uncertainty with respect to their skills and powerlessness compared with routine work. 

Considering that many studies measured stress and psychological distress at the beginning of the pandemic, compared with our data, which reflected the situation 1.5 years later, whether medical personnel and the nursing staff develop symptomatology along different lines should be examined, considering their specific roles. The interviews revealed less fear of contagion following the vaccination campaign, which was, however, accompanied by the difficult management of emotional responses of health care staff who provided care to no-vax patients. Thus, the impact of this experience on the perceived stress burden of doctors and nurses should be studied. 

Similar to the experiences of staff who cared for COVID-19 patients, HCWs who are involved in the frontline care of SARS, MERS, and EVD patients were afraid of contracting the virus or spreading it to their loved ones and were aware that they were risking their safety because of this involvement [56]. The responses of our sample emphasize the emotional experience with regard to the vaccination status of the patients and the representation of the vaccine as a safety tool for their health. Based on these considerations, future studies should examine how vaccination influences stress in HCWs. 

Several studies have evaluated traumatic outcomes of the COVID-19 pandemic in the general population [24,26,27,28], reporting cases of PTSD and anxiety/depressive symptoms in health care personnel [57], consistent with findings from studies on HCWs in emergency settings prior to the COVID pandemic [58,59]. Overall, 37% of our sample reported a high and clinically significant level of post-traumatic symptoms, which is in the upper range compared with international findings of the prevalence of PTSD in intensive care workers—40% in England [46], between 16% and 27% in France [60,61,62], and 29% in Australia [63]. Considering even mild post-traumatic symptoms, however, 73% of Canadian HCWs were found to be affected [64], consistent with the 74% rate in our sample. 

This prevalence is higher than in a survey of nurses who are considered to be highly qualified and resilient in Wuhan, China: 5.6% still reported a clinically significant level of PTSD symptoms and 22.2% scored above the clinical threshold [65]. The focus on HCWs in the intensive care unit has also revealed high levels of stress [29,30,31] and PTSD symptoms at clinically significant levels [32], even in the long term, although the incidence of this disorder decreases over time [33]. 

Overall, our results at 1.5 years after the start of the pandemic show a higher prevalence than other studies on PTSD in HCWs in ICUs, which were conducted at its outset. These data thus argue against the hypothesis that post-traumatic symptomatology wanes over time [33], necessitating further investigation by taking into account the complexity of comparing different contexts. 

Despite the negative traumatic impact of the frontline experience on HCWs [66,67], the positive responses triggered by this experience should be considered [68]. To this end, PTG outcomes in HCWs who are engaged in the pandemic response have been examined extensively [35,67,69,70,71,72,73,74]. Consistent with these studies, our survey showed greater PTG in the ‘Appreciation of Life’ dimension—i.e., in the redefinition of priorities and life value following the event—and in the ‘New Possibilities’ dimension, at the expense of ‘Spiritual Change’, which was the least represented dimension. However, in contrast to some surveys [35,67,73], 54% of our sample scored at or below the average of the normative sample in the “Personal Strength” dimension; thus, despite its presence, it highlights the perceived lack of growth of health professionals. In our sample, the “Relating to Others” dimension was not significant, in contrast to other studies [35,67] in HCWs who care for COVID-19 patients, perhaps because the study by Feingold and colleagues was conducted during the first wave of the pandemic, in 2020, when levels of social attention and consideration toward HCWs were high. We speculate that the initial state of fear and uncertainty, combined with the media and social impact, influenced growth with respect to the perception of solidarity and communal closeness, and that it decreased in the long term. 

Differently from the above mentioned studies, in our research, the analysis of PTG was not limited to being only quantitative. In fact, through open-ended questions aimed at investigating the positive and negative factors of the experience of caring for COVID patients, we also qualitatively described the factors that HCWs thought were involved in defining their PTG. The three macro-areas identified in the analysis—sense of emotional-relational competence, sense of clinical-technical competence, and quality of relations in the workplace—allowed us to categorize the experiences of the HCWs with respect to regression or growth as a consequence. The three domains we have circumscribed and the two dimensions of the PTGI show that the sudden demand to care for patients in stressful circumstances, because of the pandemic and intensive care, might have strengthened the internal resources of caregivers, giving rise to PTG. 

The relationship between post-traumatic symptomatology and PTG is notable and debated. Certain authors [75] suggest that there is no correlation between the two phenomena, whereas others [76] believe they are positively correlated. Others [76,77] point out the possible linear and curvilinear relationship between PTG and trauma symptomatology. In the period before COVID-19, moderate levels of post-traumatic symptoms were associated with higher levels of PTG among HCWs [78]. The author identified this relationship in psychologists and not in nurses. 

Consistent with others [79,80], our results highlight a correlation between PTGI and IES, suggesting a positive relationship between the psychological impact of the event and possible post-traumatic growth.

Based on this perspective, post-traumatic symptomatology emerges as the pathway through which PTG can occur. For there to be PTG, post-traumatic stress must be present, allowing an individual to implement the appropriate and necessary strategies to cope with it, but it must not exceed a threshold that results in inhibition of the individual [35,36]. The literature suggests that HCWs that provide first-line care and treatment of COVID-19 patients are at greater risk than those in second-line care [58,59]. Further, HCWs in the ICU have higher levels of stress than workers in other units [29,30,31,81].

We hypothesize that work patterns, prolonged over time and embedded in a landscape of uncertainty and pressure, contributed to chronic stress in HCWs on the front-line. Such repeated stressors oppose what is experienced by patients who are admitted to the ICU for COVID-19, examined in our previous study [82] during the same period and in the same clinical context. Patients who were admitted to the same ward in fact reported higher levels of PTG and substantially lower levels of anxiety, depression and post-traumatic symptoms. The patients experienced a condition of extreme helplessness, for a limited period of time with a consistent fear of dying; the HCWs, on the other hand, experienced a condition of fear for their health and the emergency context in a chronic form that lasts over time. Considering the evident differences in the role and experience of the two samples, we hypothesize that the subjective experience of helplessness, which declined in disparate manners between samples, played a primary role.

In conclusion, the results of this study indicate that a significant proportion of HCWs showed intermediate to high anxious/depressive symptoms (45%) and clinically significant levels of post-traumatic symptoms (37%). At the same time, post-traumatic-growth-related dimensions of “Appreciation of Life” and “New Possibilities” showed consistent resiliency in our sample. Consistent with the quantitative analysis, the qualitative analysis also found an experience of growth in the quality of relationships in the workplace and in emotional-relational and clinical-technical competence. This study also reported a block experience related to the difficulty of acquisition, regression, and loss of HCWs’ skills.

### 4.1. Limitations 

This study has several limitations, related primarily to the composition of the sample. In addition to the absence of a control sample and the small sample size, for contextual reasons, it was not possible to compare the experience of health care personnel who worked on the front-line with those who continued to work at other levels of the pandemic. Moreover, the measurement at a single time point imposed by the context prevented us from testing interpretative hypotheses with respect to the evolution of psychological distress symptomatology in a chronic situation over time.

### 4.2. Future Perspectives

This study creates several possibilities for future investigation. It will be interesting to examine the qualitative experiences of HCWs during service in the COVID ward to better understand the results on PTG. Further, we would like to study the causes of the various effects of psychological and perceived stress between medical and nursing staff. 

To complement the qualitative analysis, the implications of vaccination on the experiences of HCWs who care for COVID-19 patients should be determined. 

Finally, we would like to perform a longitudinal study of the impact of front-line experience in treating COVID-19 on the mental health of HCWs. 

## Figures and Tables

**Figure 1 healthcare-10-02201-f001:**
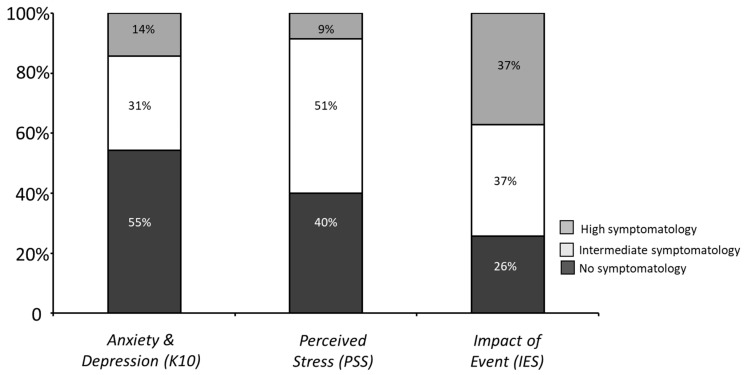
Frequencies of HCWs showing high, intermediate, and low symptomatology on the K10, PSS, and IES (anxious/depressive symptoms, perceived stress, and PTSD symptoms).

**Figure 2 healthcare-10-02201-f002:**
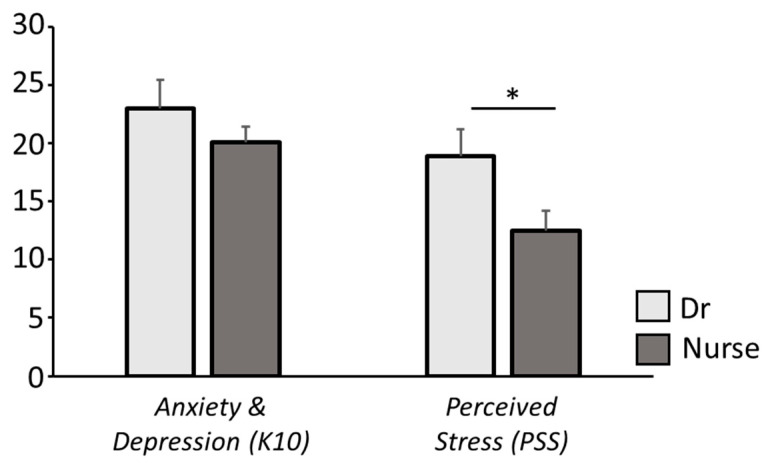
Significant lower perceived stress levels were observed in nurses compared with medical doctors. * *p* < 0.05.

**Figure 3 healthcare-10-02201-f003:**
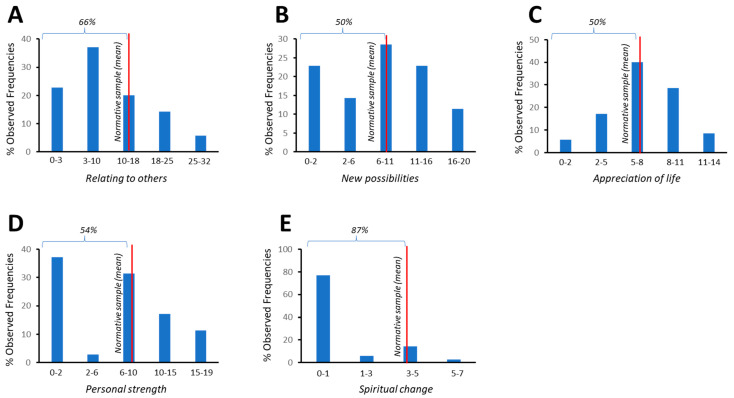
The fraction of patients who scored (% observed frequencies) at or below the normative reference sample (red line) was as follows: 66% for “relating to others”, 50% for “new possibilities”, 50% for “appreciation of life”, 54% for personal strength, and 87% for “spiritual change; subscales (**A**–**E**).

**Table 1 healthcare-10-02201-t001:** Descriptive statistics of the IES-R and PTGI subscales (mean + st.dev.).

Measure	M	SD
**PTGI**—Relating to others	10.60	8.50
**PTGI**—New possibilities	8.46	5.85
**PTGI**—Personal Strength	7.11	5.74
**PTGI**—Spiritual Change	1.03	1.89
**PTGI**—Appreciation of Life	7.31	3.48
**IES**—Avoidance	1.25	0.85
**IES**—Intrusiveness	1.53	0.85
**IES**—Iperarousal	1.19	0.88

**Table 2 healthcare-10-02201-t002:** Associations between K10, PSS, IES, and PTGI scores in all sample.

		K10	PSS	IES				PTGI				
				AV	IN	IP	TS	RO	NP	PS	SC	AL
K10		1.00										
PSS		**0.76**	1.00									
IES	Avoidance (AV)	**0.47**	0.23	1.00								
	Intrusiveness (IN)	**0.55**	0.33	**0.75**	1.00							
	Iperarousal (IP)	**0.73**	**0.57**	**0.65**	**0.75**	1.00						
	Total Score (TS)	**0.63**	**0.38**	**0.89**	**0.93**	**0.87**	1.00					
PTGI	Relating to others (RO)	0.14	−0.18	0.12	0.22	0.01	0.15	1.00				
	New possibilities (NP)	0.18	0.04	**0.43**	**0.51**	**0.40**	**0.51**	**0.78**	1.00			
	Personal Strength (PS)	0.12	0.03	0.31	**0.47**	0.27	0.40	**0.82**	**0.87**	1.00		
	Spiritual Change (SC)	0.30	−0.06	0.25	0.27	0.22	0.29	**0.33**	**0.42**	0.31	1.00	
	Appreciation of Life (AL)	0.18	0.06	0.31	**0.51**	**0.37**	**0.45**	**0.61**	**0.75**	**0.70**	**0.42**	1.00

bold = significant correlations.

## Data Availability

The data presented in this study are available on request from the corresponding author.

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
