# Peer review of "Psychological Health in Intensive Care Unit Health Care Workers after the COVID-19 Pandemic"

_healthcare, 2022, doi:10.3390/healthcare10112201_

Round 1
Reviewer 1 Report
Although the COVID-19 pandemic had an impact on the general population, health care workers (HCWs) constituted one of the groups that were most adversely affected by the associated risks, due to the significant consequences on their mental health.
Authors examined these psychological effects on HCWs who cared for COVID-19 patients who were admitted to the intensive care unit in an Italian hospital.
Subjects were administered several self-reported questionnaires: Kessler 10 Psychological Distress Scale (K10), Perceived Stress Scale-10 (PSS), Impact of Event Scale Revised (IES-R) and Post-traumatic Growth Inventory (PTGI), and 2 open-ended questions that were oriented toward understanding their emotional experience. The first open-ended question addressed the positive and negative aspects of their experience, and the second examined differences in the experience of professionals between the first and second phases of the emergency.
Their results showed in brief: (a) overall, 45% of HCWs showed medium-to-high anxiety-depressive symptoms, whereas 60% presented with medium-to-high levels of perceived stress. (b) Nurses experienced significantly lower PSS scores than physicians.(c) In addition, 37% of subjects developed symptoms of PTSD, and 50% showed post-traumatic growth in the "appreciation of life" and "new possibilities" dimensions.
The authors identified with regard to the open-ended questions, 3 themes were identified: quality of workplace relationships, sense of emotional-relational competence, and sense of clinical-technical competence.
In addition, the authors identified the 3 areas that were common to the responses made it possible to identify 2 macro categories of responses: Growth and Block.
Authors concluded that: (I) the mental health of HCWs who are involved in the front line of COVID-19 was significantly impacted by this experience, showing high levels of post-traumatic stress and anxiety and depressive symptoms more than 1 year after the emergency began. (II) A qualitative analysis of staff experiences can be a useful guide for structuring interventions and prevention.
This is an attractive, interesting and well written paper.
I have some minor suggestions with a pure academic spirit:
1. Abstract: better summarize proportionally the sections and avoid the use of our
2. In the introduction there are the hypothesis, however a clear purpose is lacking
3. Insert a table with acronyms
4. Figures must be improved (the resolution is low)
5. Par. 3.1 and sub pars are not clear and confused
6. Insert the conclusions with the achievements of your study
7. Check the MDPI standard for the body of the manuscript, there are several points to fix
Author Response
This is an attractive, interesting, and well written paper.
I have some minor suggestions with a pure academic spirit.
- Abstract: better summarize proportionally the sections and avoid the use of our
#1. Author’s response. The abstract has been modified as requested.
- In the introduction there are the hypothesis, however a clear purpose is lacking
#2. Author’s response. A clear purpose of the study has been now included in the manuscript (lines 97-101)
- Insert a table with acronyms
#3. Author’s response. A list of acronyms is now included in the article (lines 50-54)
- Figures must be improved (the resolution is low)
#4. Author’s response. High resolution Figures will be uploaded with the new version of the manuscript.
- 3.1 and sub pars are not clear and confused
#5. Author’s response. We checked the clarity of this section. The format used is similar to the one used in other published papers (e.g. doi 10.3389/fpubh.2022.951136)
- Insert the conclusions with the achievements of your study.
#6. Author’s response A conclusive paragraph has been added to the article (lines 595-603)
- Check the MDPI standard for the body of the manuscript, there are several points to fix
#7. Author’s response. We checked the MDPI standards.
Reviewer 2 Report
This is an important research topic and the authors have provided a well-written manuscript to describe the measures assessed and their methodology. There is an obvious limitation of the small sample size, but given the specialized nature of the sample, and the mixed-methods approach, it seems worth looking beyond.
The major limitation that does need to be addressed is a discussed of the mixed-method approach to data collection and an integration of the data. Frequency data is somewhat helpful, but it would be more useful to give an idea of how the quantitative and qualitative data support one another. I recommend adding a correlation matrix to see how the continuous quantitative variables relate as well as a table that integrates the quantitative data with the qualitative data. This would then lead to a re-write of the discussion section.
Without additional analyses, I'm not sure that demonstrating that frequency scores on these measures being higher than normative samples is that worthy of publication.
Author Response
The major limitation that does need to be addressed is a discussed of the mixed-method approach to data collection and an integration of the data. Frequency data is somewhat helpful, but it would be more useful to give an idea of how the quantitative and qualitative data support one another. I recommend adding a correlation matrix to see how the continuous quantitative variables relate as well as a table that integrates the quantitative data with the qualitative data. This would then lead to a rewrite of the discussion section. Without additional analyses, I'm not sure that demonstrating that frequency scores on these measures being higher than normative samples is that worthy of publication.
#1. Author’s response. We edited our paper taking into account the referee’s comments. He pointed out to us the need to clarify why we decided to use a combined approach of qualitative and quantitative analysis of the investigated phenomenon. In the current version of the manuscript, we describe the protocol used and the sequentiality of the chosen procedure. In particular, we specified that an initial quantitative approach was followed by a qualitative approach that allowed us to analyze in detail the PTG construct in our sample. In fact, unlike previous studies in which the analysis of PTG was limited to being quantitative, here, through open-ended questions aimed at investigating the positive and negative factors of the experience of caring for COVID patients, we also investigated the elements that HCWs thought were involved in their PTG.
This is now made explicit in our article, in the methods (lines 157-159), results (lines 215-224; lines 230-232), and discussion (lines 430-434) sections. In addition, we have added a correlation table between the quantitative data obtained from the different questionnaires and modified the discussion as requested by the referee (Table 2).